# Economic Evaluation of Anesthesiology-Led Cardiac Implantable Electronic Device Service

**DOI:** 10.3390/healthcare11131864

**Published:** 2023-06-27

**Authors:** Ahmed Zaky, Ryan L. Melvin, David Benz, James Davies, Vardas Panayotis, William Maddox, Ruchit Shah, Tom Lynch, Adam Beck, Kristine Hearld, Tom McElderry, Miriam Treggiari

**Affiliations:** 1Department of Anesthesiology and Perioperative Medicine, University of Alabama at Birmingham, Birmingham, AL 35205, USA; rmelvin@uabmc.edu (R.L.M.); dbenz@uabmc.edu (D.B.); tlynch@uab.edu (T.L.); 2Department of Surgery, Division of Cardiothoracic Surgery, University of Alabama at Birmingham, Birmingham, AL 35205, USA; jdavies@uabmc.edu (J.D.); pvardas@uabmc.edu (V.P.); 3Department of Cardiology, Division of Electrophysiology, University of Alabama at Birmingham, Birmingham, AL 35249, USA; wmaddox@uabmc.edu (W.M.); ruchitshah@uabmc.edu (R.S.); tommcelderry@uabmc.edu (T.M.); 4Department of Surgery, Division of Vascular Surgery and Endovascular Therapy, University of Alabama at Birmingham, Birmingham, AL 35249, USA; awbeck@uabmc.edu; 5School of Health Professionals, University of Alabama at Birmingham, Birmingham, AL 35233, USA; khearld@uab.edu; 6Department of Anesthesiology and Perioperative Medicine, Duke University, Durham, NC 27708, USA; miriam.treggiari@duke.edu

**Keywords:** cardiac implantable electronic devices, cost reduction, cost analysis

## Abstract

Background: Implementation of an anesthesiology-led cardiac implantable electronic device (CIED) service can be viewed to have economic and efficiency challenges. This study evaluates the cost savings of an anesthesiology-led CIED service. Methods: A total of 830 patients presented in the pre-implementation period from 1 March 2016 to 31 December 2017, and 1981 patients presented in the post-implementation period from 1 January 2018 to 31 October 2021. Interrupted time-series analysis for single-group comparisons was used to evaluate the cost savings resulting from reduction in operating room (OR) start delays for patients with CIEDs. Results: OR start-time delay was reduced by 10.6 min (95%CI: −20.5 to −0.83), comparing pre- to post-implementation. For an OR cost of USD 45/min, we estimated the direct cost to the department to be USD 1.68/min. The intervention translated into a total cost reduction during the intervention period of USD 250,000 (USD 18,000 to USD 470,000) per year for the institution and USD 9800 (USD 730 to USD 17,000) per year for the department. The yearly cost of employing a full-time team of CIED specialists would have been USD 135,456. The service triggered electrophysiology consultation on 13 device malfunctions. Conclusions: An anesthesiology-led CIED service resulted in substantial cost savings, increased OR efficiency and patient safety.

## 1. Introduction

The number of patients needing cardiac implantable electronic devices (CIED) and presenting for surgery is on the rise [1,2]. With advancement in medical care and increased longevity, more patients with CIEDs are undergoing surgical procedures.

The advancement of device technology and the propensity for complications occurring in the perioperative period have created a need for special training and guidelines for the management of CIEDs perioperatively. The Heart Rhythm Society, the American Society of Anesthesiologists and the American Heart Association have all published several practice advisories on perioperative management of CIEDs [3,4].

Failure to adhere to expert consensus statements of perioperative interrogation and programming of CIEDs may create an economic burden to institutions resulting from delays of elective cases awaiting the arrival of providers especially trained in the management of these devices [5]. Additionally, patient safety concerns can occur, such as cardiac asystole resulting from failure to apply an asynchronous mode to permanent pacemakers in pacing-dependent patients, and fatal arrhythmias resulting from failure to restore anti-tachycardia therapy in defibrillators to their preoperative active status [6]. Staff shortages and overwhelmingly busy schedules limit the timely availability of cardiologists and electrophysiologists to manage CIED perioperatively. Additionally, troubleshooting of complex device malfunctions may be beyond the scope of device vendor representatives who are commonly asked to manage CIEDs perioperatively [7]. All this has created a need for anesthesiologists to gain the knowledge and skills needed to safely and efficiently manage CIEDs perioperatively.

There is a paucity in studies evaluating the economic impact of anesthesiology-led CIED services particularly in the era of enhanced device technology [8,9,10].

Here we report our initial experience in building an anesthesiology-led CIED service, and we hypothesize that such a service is associated with cost savings resulting from reductions in operating room delays, enhancement of patient safety due to reductions in preventable CIED-related complications and a reduction in the length of postoperative stay (LOS).

## 2. Materials and Methods

### 2.1. Study Design

This is a single-center retrospective cohort study conducted at a tertiary care academic center to evaluate the cost savings of an anesthesiology-led CIED service. A pilot phase of the study commenced on 1 February 2017 and ended on 31 December 2017, in which the team leader interrogated approximately 20% of patients with CIEDs under the supervision of cardiac electrophysiology (EP) services. Since the pilot phase was conducted under the direction of EP services, it was included in the pre-implementation phase. Data were collected between 1 March 2016 and 31 December 2017 (pre-implementation) and from 1 January 2018 to 31 October 2021 (post-implementation)**.** The study was approved by the University of Alabama at Birmingham (UAB) Institutional Review Board (IRB) with a waiver of informed consent.

### 2.2. Outcomes

The primary endpoint was cost savings resulting from the reduction in OR case delays for any patient with CIEDs; this was defined as the difference between the actual in-operating room time, which is documented routinely for each surgery, and the scheduled in-room time. Secondary endpoint was postoperative LOS in the form of time in hours spent in the post-anesthesia care unit (PACU) and intensive care unit (ICU).

### 2.3. Establishment of Anesthesiology-Led CIED Service

The Departments of Anesthesiology & Perioperative Medicine and Cardiology along with the Division of Electrophysiology at UAB initiated a quality initiative to address operating room delays and patient safety concerns created by the overwhelming increase in the volume of patients with CIEDs undergoing cardiac and non-cardiac procedures. Beginning in February 2017, a decision was made to train and credential a selected group of anesthesiologists with fellowship training in either or both cardiac anesthesiology and critical care medicine. Prior to this decision, the CIED service was run by device vendor representatives under the supervision of the Division of Cardiology.

The project plan was to train the service leader who would initially lead a hybrid team of practitioners in the form of cardiology fellows and device representatives who would incrementally be replaced by anesthesiologists. The team leader on the Anesthesiology lead CIED service (AZ) volunteered to receive formal advanced training at the Arrhythmia Technology Institute for 3 weeks. This was followed by supervised training and management of CIEDs at the EP Device Clinic under the direction of the chief of the Section of Cardiac Electrophysiology (TM) at UAB until a total of 200 CIEDs were interrogated and programmed by the team leader at the Device Clinic between February 2017 and June 2017. The team leader voluntarily attended the Device Clinic one day a week on his post-call days. A credentialing letter was then written by the Chief of the Section of Cardiac Electrophysiology allowing the team leader to function autonomously with ‘perioperative CIED programming and interrogation’ (Appendix A). We then designed and developed a joint protocol for perioperative CIED management by an anesthesiology-led CIED service and the Division of Cardiac Electrophysiology in accordance with published HRS guidelines (Appendix B). The perioperative protocol detailed guidance on the following steps: a. Baseline settings and dates of last interrogation of CIEDs; b. Determination of pacer dependency; c. Testing for battery longevity, sensing, capture thresholds and leads impedances; d. Programming for specific procedures in terms of turning to asynchronous mode, disabling of rate responsiveness, disabling of anti-tachycardia therapy for ICDs, application of magnets and individualized programming for patients with cardiac resynchronization therapy (CRT) undergoing left ventricular assist device implantation in a form disabling left ventricular pacing and pursuing atrial based pacing and lower rate limit adjustments and e. Specific adjustments of newer CIED technology in the form of subcutaneous ICDs and leadless pacemakers, and the restoration of CIED to baseline settings (including individualized adjustments for patients with LVADs) (Figure 1) The manufacturers of 4 CIED devices supplied the programmers and equipment for CIED interrogation free of charge. On 1 January 2018, the service started between 7 a.m.–5 p.m. An official hand-off sheet between the anesthesiology team leader and on call cardiology fellows was created for surgical procedures extending after working hours and when the team leader was not available. According to the credentialing letter, liability for deviation from the CIED management protocol was borne by the team member who performed the initial interrogation procedure.

### 2.4. Study Population

Adult patients undergoing elective and emergent surgical procedures at UAB Main Hospital between 1 March 2016 and 31 December 2017 (pre-implementation) and from 1 January 2018 until 31 October 2021 (post-implementation). Patients younger than 18 years, and those scheduled for magnetic resonance imaging and radiation oncology suites were excluded from the study. Multiple and redo surgeries were counted as new procedures both pre- and post-implementation of the service.

### 2.5. Data Collection

Electronic medical records (EMR) were abstracted for the duration of the study. The International Classification of Diseases, Ninth Edition (ICD-9) was used to retrieve information on patients with ‘permanent pacemakers’, implantable cardioverter defibrillators and cardiac resynchronization therapy (CRT). Information on surgical procedures, demographics and clinical characteristics were retrieved. Pre- and post-postoperative notes written by the anesthesiology-led CIED service team were identified from the EMR for the study period. Data for expected case start times, actual case start times, postoperative LOS, total hours in PACU or ICU, total hours in stepdown unit, total hours in the general care floor and the presence of a CIED for each given day were analyzed for all hospital locations for elective surgical procedures from March 2016 to October 2021 at UAB.

### 2.6. Economic Evaluation

The cost of operating room (OR) delays was estimated to be USD 45/min by the UAB Director of Anesthesia Services and the Department of Anesthesiology Executive Administrator. Using the work of Ellis et al. [9], the cost of anesthesiologist labor due to delay is estimated to be USD 1.68/min, which represents the direct cost to the Department of Anesthesiology and Perioperative Medicine.

For estimating the cost of a similar service with a dedicated team of four advanced care practitioners (ACP), our department’s executive administrator provided a total full-time position cost estimate of USD 135,456 per year. The number of full-time ACPs was determined based on consultations from institutions with a comparable volume of CIEDs that have endorsed this CIED service model.

### 2.7. Statistical Analysis

Due to the large number of potential confounders for OR case delays, we chose interrupted time series (ITS) analysis [11,12,13,14] to assess whether the start of the perioperative CIED service was associated with intercept or slope change in the time series of OR cases delays in minutes. The ITS process described by Bernal et al. in 2017 [15] was utilized for our analysis. Broadly speaking, ITS is a modeling technique used to examine whether an underlying trend in a regularly spaced sequence of measurements of a particular value of interest (a time series) is disrupted (or “interrupted”) by an event or intervention a specific point in time. A model is built assuming that the trend was disrupted and is then compared to a null model in which the trend is not changed by the intervention. The null model provides counterfactual predictions, which estimate the time series’ values post-intervention if the intervention had not occurred.

The assumption made to address confounders with ITS analysis is that potential confounders and biases were symmetrical across the specific dates of the intervention. Therefore, the method comes with the inherent limitation that any confounders or biasing factors not meeting this assumption are not adequately controlled for by ITS.

We carried out the ITS process for both sets of observations for all cases involving a patient with a CIED, contrasting the hospital locations before and after CIED service implementation for elective surgical procedures. We could not reasonably identify a comparison group for all cases’ delays, as case end times for patients with CIEDs may impact case start times for patients without CIEDs (and vice versa) [16]. That is, we could not determine an independent control group for this quasi-experimental analysis.

We consider a linear model of the form:$$Y_{t}=\beta_{0}+\beta_{1}T+\beta_{2}X_{t}+\beta_{3}TX_{t}+\epsilon_{t}$$
where Y is the average case delay minutes by month for cases involving patients with CIEDs, T is a count of months since 1 March 2016 and X is an indicator variable that is 0 before 1 January 2018 and 1 during and afterward. $\beta_{0}$ is the intercept (baseline level at the start of reporting—March 2016), $\beta_{1}$ is the change associated with a 1-month increase in time (i.e., slope), $\beta_{2}$ is the change in intercept after the start of the CIED service and $\beta_{3}$ is the change in slope after the start of the CIED service. In sensitivity analyses, additional models that considered the announcement of the service start as an intervention were also fitted.

For all model forms, we first fit an ordinary least squares (OLS) model to the reported data. We then used a Durbin–Watson test augmented by visual inspection of model residuals as a function of time, auto-correlation function (ACF) plots and partial ACF (PACF) plots to investigate the need for inclusion of autoregressive (AR) and/or moving average (MA) components in the correlation structure of the model. For such models, fitting to reported data was performed by maximizing log likelihood.

By comparing the counterfactual estimates of average monthly case delay minutes to those from these models (Figure 2), the absolute and relative impact of the change in the time series were calculated at the beginning and end of the post-intervention window.

For the analyses of postoperative LOS, total hours in PACU or ICU, total hours in stepdown unit and total hours in the general care floor, we applied these same methods on each of these time series.

In sensitivity analyses, additional models that considered the announcement of the service start as an intervention were also fitted.

Correction for multiple comparisons was performed using Hommel’s method [17], as implemented in the Python package (Table 1) [18].

## 3. Results

### 3.1. Study Population

A total of 830 patients with CIEDs were identified in the pre-intervention cohort and 1981 patients were identified in the post-intervention group. The groups were comparable in age and comorbidities. Patients in the post-intervention group underwent more surgical procedures (*p* = 0.014) compared with the pre-implementation group.

#### All CIED Case Delays

The sample of all cases where a patient had a CIED during the study period included 2811 patients: 830 in the baseline period and 1981 in the intervention period (Table 1).

For the OLS model fitted to average delays of all CIED cases, the slope change estimate ($\beta_{3}$) was not statistically significant (*p* > 0.05), but the estimate for the intercept change ($\beta_{2}$) was (*p* < 0.05), indicating that a change in the level (or intercept) of the underlying trend of the time series of average monthly case delay minutes for CIED patients was associated with the initiation of the perioperative CIED service.

By comparing the counterfactual estimates (using only the estimates for $\beta_{0}$ and $\beta_{1}$) of average monthly case delay minutes to those from this model (using the estimates for $\beta_{0}$, $\beta_{2}$ and $\beta_{3}$), the absolute and relative impact of the change in the time series were calculated at the beginning and end of the post-intervention observation window. Because the significant change is to the intercept only, the estimated absolute change at any time point post-intervention period was −10.64 min (95%CI: −20.5, −0.83) (Figure 2).

The implementation date for anesthesia-led cardiac implantable electronic device service is associated with a statistically significant decrease of 11 min in monthly average case delay minutes for patients presenting with these devices in the operating rooms.

### 3.2. Secondary Endpoints

We found no significant change associated with the start of this service in postoperative LOS.

### 3.3. Economic Evaluation

Using the institutional cost estimate of USD 45 per OR delay minute and the work of Ellis et al. [9], we estimate the direct cost to the department at USD 1.68/min. Taking this reported average at face value, without accounting for inflation, indicates a reduction in cost of USD 480 per OR case involving a patient with a CIED (with a 95% confidence interval of USD 37 to USD 920) for the institution and USD 18 (USD 1.4 to USD 34) for the department. Between the implementation of the service and October 2021, 1981 procedures in patients with an implanted CIED were carried out at our institution, inducing a total cost reduction of USD 250,000 (USD 18,000 to USD 470,000) per year for the institution and USD 9800 (USD 730 to USD 17,000) per year for the department. These cost savings translate into time savings of approximately 5555 min per year (i.e., USD 250,000 divided by USD 45). These numbers also roughly equate to 581 cases per year on average (i.e., USD 250,000 divided by USD 480) or about 9.6 min per case (i.e., 5555.55 min divided by 581 cases).

The statistically significant estimate of the intercept change coefficient as a negative value for the post-implementation group indicates there was an immediate reduction in average monthly case delays for patients presenting with CIEDs. This reduction was sustained (not increasing or decreasing), as indicated by the non-significant estimate for the post-intervention slope change coefficient for post-implementation group (Figure 2).

For comparison, the cost of employing a full-time team of four CIED specialists would have been USD 135,456 per year. Therefore, the overall cost benefit of staff in such a position would be an expected saving of USD 115,000 per year (estimated savings of CIED service minus expected personnel costs) versus the estimated savings of the service investigated here, which used additional personnel, of USD 250,000 per year.

### 3.4. Patient Safety

During the implementation period, 13 device malfunctions were exclusively discovered by the anesthesiology-led CIED service team (Table A1). When needed, an immediate electrophysiology consult was placed to correct lead placements or to upgrade the CIEDs device based on changes in pacemaker dependency during the perioperative period. These near misses were all successfully attended to by the EP team.

## 4. Discussion

In this study, we demonstrated significant and sustained cost savings by reducing all case operating room delays for patients with CIEDs and enhancement of patient safety by an anesthesiology-led CIED service in a tertiary care academic center.

The contribution of anesthesiologists to hospital cost savings and enhancement of patient safety identifies them as key players in value-based Medicare reimbursement and provides leverage to their fair fight to increase reimbursement by public insurers compared to other specialties [19,20].

Few studies have investigated the economic impact of anesthesiology-led CIED service [9,21]. Ellis et al. [9] demonstrated a reduction in first case delay of 16.7 min post-intervention from pre-intervention in the CIED cohorts. This has translated to cost savings of USD 14,102 annually or USD 96.04 per patient with CIED. The authors have reported three adverse events in the pre-implementation period and none post-implementation. Compared to the study by Ellis et al., we used our institution rates of cost delays, studied a longer post-implementation phase, studied a sicker cohort with a significantly larger volume of cardiac and non-cardiac procedures and studied more technologically advanced CIEDs, such as leadless pacemakers [22] and subcutaneous ICDs [23]. Our statistical model allowed us to realize a sustained effect of cost savings. Furthermore, the team leader on the service underwent a different pathway of external training on a volunteering basis and attended the pacemaker clinic during non-working hours without time or monetary compensation and with no scheduling adjustments. The pre-intervention period was shorter compared to Ellis et al.’s study, as case delays were electronically tracked only a short period prior to service implementation. Distinctively, we added the cost of fulltime ACPs dedicated to performing CIED interrogation and programming. We used this rationale as this has been the service model of several institutions across the nation. Given these encouraging results, the institution is in the process of credentialing more anesthesiologists towards an all-anesthesiologists CIED service. We reported comparatively more preventable adverse events in the post-implementation period which may be due to sicker patients, longer follow up period, larger volume of cases and most importantly, adequate documentation of CIEDs interrogations. Attention to these near misses avoided potential adverse outcomes, which is an intangible yet important value added by this service. The study by Navas et al. [21], while demonstrating a significant difference in operating room delays due to CIED- related issues compared with non CIED-related issues, did not inform on the structure or actual economic impact of a formal CIED service in their institution.

Even though we found a sustained cost-saving effect of our service, our cost savings seem to be on the conservative side compared to other studies [9]. For example, the opportunity cost in reducing operative delays is the availability of the OR to do more elective/emergency surgeries in addition to anesthesiologists being available to render other revenue-producing services. The time savings is 5555.55 min per year roughly translates to about 3.85 additional 24-h days for the OR and 7.70 additional 12-h shifts for the anesthesiologists’ team. The quantification of this data was not available at the time of writing of this manuscript. Additionally, and importantly, the comparative revenues resulting from billing for this service’s pre- and post-implementation were not available at the time of writing as there has been a change in the billing system at our institution. Added to the intangible benefits of our service is the considerable qualitative benefit of each patient (or family) spending 10.64 fewer minutes in the OR (or waiting for the OR). In summary, our service has achieved both quantitative and qualitative benefits that consolidate its value. It also triggers the search for innovative methodology to truly quantify the benefits of this and alike services.

Our service has resulted in several educational benefits. A formal curriculum for rotating residents and fellows has been created, and the increased familiarity and engagement of anesthesiologists with CIED management has resulted in several publications [24,25].

Our study suffers from some limitations. Pre-implementation data on cost delays were only collected for a short period of time as there was no formal system of tracking case delays prior to that period. However, it is likely that more cost savings would have been made had our tracked pre-implementation period been longer. We did not report adverse events in pre-implementation because of a lack of consistent documentation on CIED interrogations pre-implementation, as the service was partially managed by manufacturer representatives who were not authorized to write formal notes in the EMR. When including model terms for the announcement phase of the service, those coefficients were not statistically different from 0 (*p* > 0.05). Therefore, we reconciled the pilot phase of the service, when service was not started in full capacity, to the pre-implementation phase to simplify the analysis. It is likely that this reconciliation has underestimated operating room delays pre-implementation. Whereas other studies have only looked at first case delays, we analyzed all case delays. This is due to a higher proportion of patients with devices as next cases compared to first cases, and to a greater shortage of device reps and cardiology fellows during the later hours of the day at our institution. At the time of data collection for this manuscript only one anesthesiologist was credentialed to perform CIED interrogations with cross covering from cardiology fellows after and during absence hours. Currently there are more anesthesiologists who are being credentialed to manage CIEDs with the goal of replacing non-anesthesiologist providers by the end of 2024. The team leader on this service has borne the expense of training, was not compensated by the institution for time and effort and had no adjustments in daily case schedule to pursue CIED interrogations. Accordingly, and in contradistinction to similar services in other institutions, our institution incurred no costs in training or compensating the team leader. While this level of dedication may not be generalizable, it paved the way for a greater recognition of anesthesiologists’ value in this area and allowed more structuring of this growing CIED service. Another limitation of our study is that we did not account for provisional retention costs of anesthesiologists managing these devices and how this service has resulted in any burnouts of the team members [26]. While this may not be a prominent consideration at the present stage, it is an important consideration in the long run and will be part of our future ongoing assessment of this service.

## 5. Conclusions

In summary, an anesthesiology-led CIED service broadens the scope of anesthesiologists, reduces costs resulting from operating room delays, i efficiency of the operating room and enhances patient safety. Adoption of this service by other hospitals is encouraged.

## Figures and Tables

**Figure 1 healthcare-11-01864-f001:**
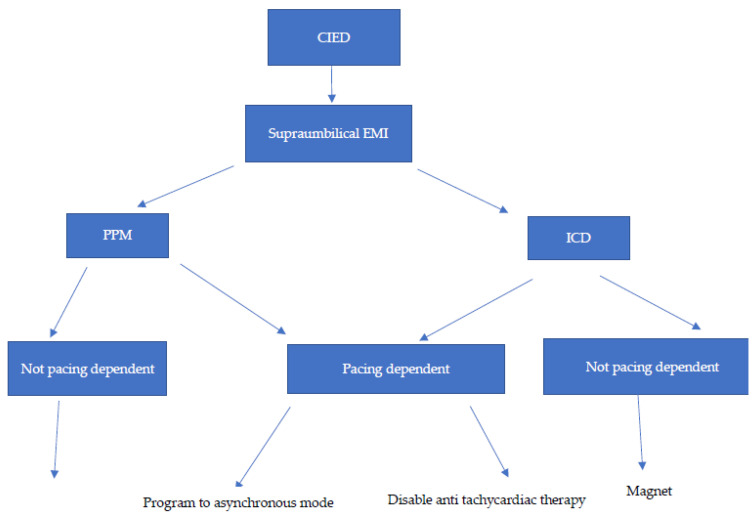
Flow chart for perioperative interrogations, programming and restoration of cardiac implantable electronic devices. CIEDS: cardiac implantable electronic devices; PPM: permanent pacemakers and ICD: implantable cardioverters defibrillators.

**Figure 2 healthcare-11-01864-f002:**
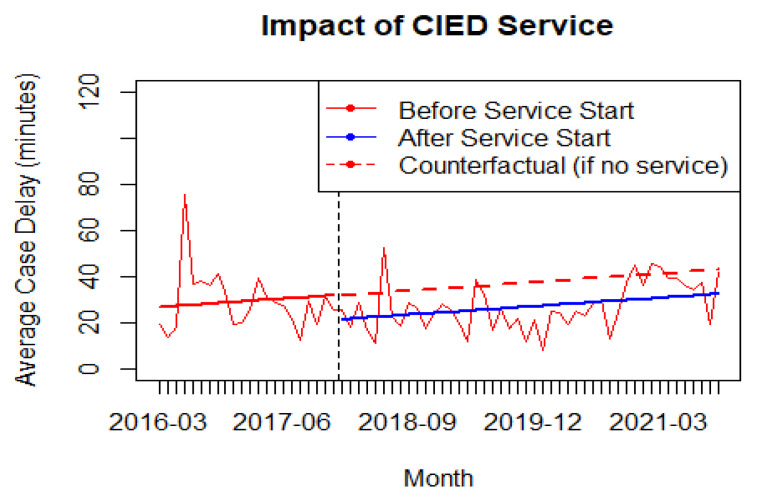
Impact of Anesthesia-led CIED service.

**Table 1 healthcare-11-01864-t001:** Characteristics of studied population.

Parameter	n = 2811		
	Pre-Intervention	Post-Intervention	*p* Value (Adjusted)
	(n = 830)	(n = 1981)	
Age (mean ± SD)	64.21 ± 15.08	64.66 ± 14.36	0.5
Surgical Specialty (n (%))			0.014
EMR-SURG	1 (0.1)	1 (0.1)	
BREAST	0 (0)	1 (0.1)	
Burns	2 (0.2)	14 (0.7)	
CARD	229 (28)	543 (27)	
THOR-Non TX	37 (4)	76 (4)	
GI	76 (9.16)	192 (9.69)	
ENT	67 (8.1)	146 (7.5)	
GYN	14 (1.7)	65 (3.3)	
LIVER TX	4 (0.5)	7 (0.4)	
NEURO	52 (6.4)	121 (6.1)	
OPTH	0 (0)	5 (0.3)	
ORAL	29 3.6)	72 (3.6)	
ORTHO	122 (15)	227 (11.46)	
PLAS	9 (1.2)	32 (1.7)	
RENAL TX	20 (2.4)	42 (2.1)	
ROBOTIC	3 (0.4)	1 (0.1)	
SURG ONC	37 (4.5)	61 (3.1)	
TRAUMA	19 (2.3)	25 (1.3)	
Lung TX	0	2 (0.1)	
URO	29 (3.5)	37 (1.9)	
VASCULAR	103 (12)	295 (15)	
Missing	4 (0.5)	13 (0.7)	
ASA Status (n (%))			0.5
1	0	4 (0.2)	
2	5 (0.6)	11 (0.6)	
3	371 (44.7)	873 (44.1)	
4	428 (51.6)	1045 (52.8)	
5	26 (3.1)	48 (2.4)	

CARD: cardiac; THOR: thoracic; ENT: ear nose and throat; GI: gastrointestinal; GYN: gynecologic; Tx: transplantation; NEURO: neurologic; OPH: ophthalmologic; ORTHO: orthopedic; URO: urologic; SURG ONC: surgical oncology; PLAS: plastic; and ASA: American Society of Anesthesiologists. Correction for multiple comparisons was performed using Hommel’s method [17], as implemented in the Python package [18].

## Data Availability

The data that support the findings of this study are available on request from the corresponding author. The data are not publicly available due to privacy or ethical restrictions.

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
