# Peer review of "Economic Evaluation of Anesthesiology-Led Cardiac Implantable Electronic Device Service"

_healthcare, 2023, doi:10.3390/healthcare11131864_

Round 1

Reviewer 1 Report

Background:

In this study, the authors conduct an economic evaluation of providing an anesthesiology-led cardiac implantable electronic device (CIED) service. There is a clear need for knowledgeable individuals for the perioperative management of the CIED devices. More than one study advocates that the anesthesia team is best suited for this purpose (refer Schulman & Rozner, 2013; Rooke & Bowdle, 2013; Izrailtyan et al., 2013; Thompson & Mahajan, 2013; and others). Using the data from a single evaluation center, the authors conduct an economic evaluation of the anesthesiology-led CIED service. Overall, the authors have done a satisfactory job at addressing an important topic of the economic evaluation of anesthesiology-led CIED service, though there is some scope for improvement.

Feedbacks:
1) Why anesthesiologist? Though CIED service does not fully come under a physician's field of expertise, the perioperative needs of CIED service can be fulfilled by a field of expertise that encompasses a large fraction of overlapping skill sets, especially after receiving training and certifications. I agree with the author's assessment (as well as several other studies) that the anesthesiology team is generally the best suitor. But since there are no nationally-recognized benchmarks for training and certifications, and the entire training schedule was done in-house, how rigorous and encompassing the training was comes into play before making the service into a policy for the author's institution as well as a prescription for other institutions to follow. What is the author's estimate for the economic uncertainty involved in making an anesthesiology team (without inputs from the electrophysiology team) entirely responsible for perioperative decisions on the CIED. A sensitivity analysis on the economic evaluation of direct costs would allow for this uncertainty estimate to vary between zero (anesthesiology team can handle all uncertainties) to one (anesthesiology team requires the assistance of the electrophysiology team or others during all interventions). I would love to see the data, if possible, on how many occasions the trained anesthesiology team required the assistance of external experts during an intervention. This directly relates to the effectiveness of the service. For a total of 1,981 cases, the authors quote that there were only 13 adverse events (Table A3), all handled by the in-house  anesthesiology team. Does that mean that the anesthesiology team did NOT require any assistance for external experts or CIED manufacturers even once during an OR event in the post-implementation period?

2) The authors only use direct costs of the anesthesiology-led CIED service. Now, as with the introduction of any new technique (or service), a thorough economic evaluation study should consider what type of costs must be included and what type of costs can be reasonably ignored. Sufficient justification is required why costs that were included were included and why the cost that were ignored were ignored.

First off, an economic evaluation of a service must include all costs (irrespective of who incurred that cost). The study presents the direct cost of having a team of four advanced care practitioners is $135,456 per year (which roughly amounts to $33,864 per person borne by the institution). CIED training is uncompensated and no adjustments to daily case schedules were made. The first obvious limitation here is that only institutional labor costs was taken into account. Employee's (initial and on-going) training costs was not included. It would be prudent to include all costs to see if there is no externalities of uncompensated CIED training. In a similar economic analysis, Ellis et al. 2017 (paper number 10 in the reference list) included training costs. Those authors include training costs because the institution paid for the training, but by including it the analysis becomes complete.

Are there any retention and replacement costs of trained anesthesiologists? A 2021 study by Afonso et al. documents that 59.2% of anesthesiologists were at high risk of burnout, and 13.8% of anesthesiologists met criteria for burnout syndrome, one of the highest of all practitioners. Is it advisable to add to their responsibilities without reduction in other case loads? Can this cost be ignored? Also, do trained anesthesiologists leave for better prospects? Therefore, retention and replacement costs may come into play. Acknowledging this aspect in the study is also crucial. Authors can additionally address ongoing retraining costs too.

It is quite possible that training and retention costs may play a big role in long-run even though they can be ignored in short-run. The authors must address these issues either by doing a quantitative assessment of added costs in the analysis directly or by qualitatively addressing it in the discussion section.

3) The authors only use direct benefits of having anesthesiology-led CIED service. Numerically, the benefits is quoted to be about $250,000 per year (see line number 266), or $480 per OR case (see line number 262), or time savings of approximately 5,555 minutes per year (i.e., $250,000 divided by $45 estimate [line number 259]).

These number also roughly equates to 581 cases per year on average (i.e., $250,000 divided by $480), or about 9.6 minutes per case (i.e., 5555.55 minutes divided by 581 cases). These numbers are not explicitly mentioned in the study. Can the authors verify the numbers?

a) The regression estimate is time savings of 10.64 minutes per case (line number 244), which is much higher than 9.6 minutes per case (i.e., 5555.55 minutes divided by 581 cases). This regression estimate will result in much higher benefit savings (= 10.64 minutes per case * Number of actual cases done per year * $45 per minute). I would also assume that the benefit goes up over time. This also can be summarized in the paper - do the benefit in Year T, Year T+1, Year T+2, and Year T+3 based on the actual number of cases performed in each of those years.

b) The opportunity costs of operative delays is the availability of the OR to do more elective/emergency surgeries in addition to anesthesiologists being available to render other revenue-producing services. The time savings is 5,555,55 minutes per year roughly translates to about 3.85 additional 24-hour days for the OR and 7.70 additional 12-hour shifts for the anesthesiologists team. I believe those dollar amounts are readily available to boost up the benefits even further.

c) Though not a direct benefit and cannot quantitatively assessed, each patient spending 10.64 fewer minutes in the OR (or waiting for the OR) on itself is a considerable qualitative benefit that must be mentioned in the manuscript.

In sum, I believe the authors have been very conservative in documenting benefits. A more detailed working of the benefits would greatly enhance the study.

4) Table A3 provides details on the adverse events in the post-implementation period. Even though there are data limitations, I think a Table on the pre-implementation period would add more clarity to the paper, with whatever data is available.

5) The study uses ITS analysis and counts multiple and redo surgeries as new procedures in both pre- and post-implementation of the services. For the purpose of robustness, the authors should present the results/discuss the following:
(i) are there more multiple and redo surgeries in the pre-implementation period? If yes, can any of these surgeries be attributed to the delayed management of the CIED devices. And, if yes, this further strengthens the importance of their study.
(ii) To test the notion that the study's results are NOT (predominantly) driven by multiple and redo surgeries, repeat the analysis after removing the multiple and redo surgeries in both the pre- and post- periods. Now, it may very well be possible that there is a significant reduction in sample size, making it impossible to do the ITS analysis. In that case, assign a binary variable for multiple and redo surgeries in the original regression specification and repeat the analysis. 

6) Minor issue: Please provide the regression model equation in the paper. This is especially helpful for the readers when referring to the coefficients beta2, beta3 and etc in the results section. It is also not clear what other covariates were included in the regression equation. For example, what patient's demographic and health variables (age, sex, comorbidities etc) were included in the estimation. Was the coefficient estimates on these control variables similar in the pre- and post- implementation period?

Reference:
Schulman PM, Rozner MA. Anesth Analg 2013; 117:422–7. doi: https://doi.org/10.1213/ane.0b013e31829003a1
Rooke GA, Bowdle TA: Anesth Analg 2013; 117:292–4. doi: https://doi.org/10.1213/ANE.0b013e31829799f3
Izrailtyan I, Schiller RJ, Katz RI, Almasry IO. Anesth Analg 2013; 116:307–10. doi: https://doi.org/10.1213/ANE.0b013e3182768ce3
Thompson A, Mahajan A. Anesth Analg 2013;116:276–7. https://doi.org/10.1053/j.jvca.2021.04.025
Anoushka M. Afonso, M.D.; Joshua B. Cadwell, M.B.A., M.S.; Steven J. Staffa, M.S.; David Zurakowski, Ph.D.; Amy E. Vinson, M.D. Anesthesiology 2021, Vol. 134, 683–696. https://doi.org/10.1097/ALN.0000000000003722

Reviewer 2 Report

This paper focuses on evaluating the cost savings of an anesthesiology led CIED service based on data collected during pre-implementation and post-implementation.

My comments are the following:

1. The introduction of data should be present in the abstract. It is suggested that the information about data in the methods part of the abstract be separated.

2. Related studies should be argued in detail, and the novelty of this work should be discussed.

3. The estimation methodology and analysis process should be described in more detail.

4. Whether there are other methods that apply to the statistical analysis of the problem, such as DID.

5. It is necessary to introduce the dependent variables, independent variables, and control variables in the model which is estimated by OLS in detail.

6. There are some writing errors, such as in line 25.

Moderate editing of English language required.
